# The Reaction Mechanism of Loganic Acid Methyltransferase: A Molecular Dynamics Simulation and Quantum Mechanics Study

**DOI:** 10.3390/molecules28155767

**Published:** 2023-07-30

**Authors:** Mateusz Jędrzejewski, Łukasz Szeleszczuk, Dariusz Maciej Pisklak

**Affiliations:** Department of Organic and Physical Chemistry, Faculty of Pharmacy, Medical University of Warsaw, Banacha 1, 02-093 Warsaw, Poland; s078210@student.wum.edu.pl (M.J.); lukasz.szeleszczuk@wum.edu.pl (Ł.S.)

**Keywords:** methyltransferase, molecular dynamics, quantum chemical cluster approach, reaction mechanisms, enzymatic catalysis

## Abstract

In this work, the catalytic mechanism of loganic acid methyltransferase was characterized at the molecular level. This enzyme is responsible for the biosynthesis of loganin, which is a precursor for a wide range of biologically active compounds. Due to the lack of detailed knowledge about this process, the aim of this study was the analysis of the structure and activity of loganic acid methyltransferase. Using molecular dynamics (MD) simulations, the native structure of the complex was reconstructed, and the key interactions between the substrate and loganic acid methyltransferase were investigated. Subsequently, the structures obtained from the simulations were used for quantum chemical (QM) calculations. The QM calculations allowed for the exploration of the energetic aspects of the reaction and the characterization of its mechanism. The results obtained in this study suggest the existence of two patterns of interactions between loganic acid methyltransferase and the substrate. The role of residue Q38 in the binding and orientation of the substrate’s carboxyl group was also demonstrated. By employing a combined MD and QM approach, the experimental reaction barrier was reproduced, and detailed insights into the enzymatic activity mechanism of loganic acid methyltransferase were revealed.

## 1. Introduction

Among a large group of enzymes, methyltransferases possess unique properties and applications. These proteins are responsible for transferring the methyl group to the substrate molecules. The substrates for methyltransferases can be both macromolecules such as proteins [1], RNA [2] and DNA [3], as well as small organic compounds [4]. Methyltransferases are widespread among all groups of organisms, from archaea to humans, which proves their important role in processes occurring in living organisms. Mutations in the genes encoding methyltransferases are of great importance for medicine, as they can lead to diseases such as Bowen–Conradi syndrome [5] or the occurrence of excessive toxic effects of drugs, e.g., thiopurine derivatives, in people with reduced thiopurine methyltransferase activity [6].

Methyltransferases belong to the group of transferase enzymes that catalyze the transfer of functional groups between two molecules. A typical coenzyme found in methyltransferases is S-adenosylmethionine (SAM). SAM is formed as a result of enzymatic synthesis from the amino acid methionine and ATP. On the positively charged sulfur atom of SAM, there is a reactive methyl group, which can be transferred to the oxygen, nitrogen, carbon or sulfur atoms of the substrate as a result of the reaction.

One family of methyltransferases that is of particular interest for potential applications in biotechnology is the SABATH family. This family was discovered in the 1990s. Its name comes from the first discovered enzymes from this family (**sa**licylic **a**cid methyltransferase, **b**enzoic **a**cid methyltransferase and **th**eobromine methyltransferase) [7]. SABATH methyltransferases participate in the biosynthesis of secondary metabolites responsible for the smell, attracting pollinators, resistance to diseases and plant defense reactions [8]. Substrates for SABATH family methyltransferases include carboxylic acids such as salicylic, benzoic, loganic and jasmonic acids. As a result of the reaction catalyzed by the enzyme, methyl esters derivatives of carboxylic acids are formed.

One of the best-characterized SABATH family methyltransferases is loganic acid methyltransferase (LAMT). It is an enzyme involved in the biosynthesis of monoterpene indole alkaloids. A large part of this group of compounds is used in medicine, among other applications, as chemotherapeutics (vincristine and vinblastine), blood-pressure-lowering substances (ajmalicine and serpentine) and blood-glucose-lowering substances (vinzolidine). One of the stages of the formation of these compounds is the condensation of secologanin and tryptamine, which leads to the formation of isovincoside, which is a precursor of many indole alkaloids with a diverse structure [9]. 

The biosynthesis of secologanin has been described in detail for pink periwinkle (Catharanthus roseus). This process starts with the primary metabolite geranyl pyrophosphate, which is converted to geraniol derivatives, which then form nepetalactone. It undergoes oxidation, glycosidation and hydroxylation to loganic acid. The next step of biosynthesis is catalyzed by LAMT. This process is a one-step reaction and involves the transfer of a methyl group from the SAM cofactor to one of the oxygen atoms of the carboxyl group of the loganic acid. As a result, SAH and a methyl ester of loganic acid (loganin) are formed (Figure 1). Loganin is then converted into secologanin by its synthase [10].

Although LAMT-mediated methylation is a simple process, the exact mechanism of catalysis remains unknown. Some details regarding the structure and the reaction catalyzed by this enzyme were revealed by the publication of the crystal structure of LAMT from Catharanthus roseus in 2018 [11]. This structure (PDB code 6C8R) is a homodimer, and SAH is bound in both active centers of the enzyme, making it structurally similar to SAM, but lacks a reactive methyl group. Although LAMT is a homodimer, loganic acid is bound only in one of the active sites of the enzyme. This shows that the active sites of the enzyme are not equal and can catalyze the methylation reaction alternately. The methyltransferase subunits interact with each other in regions that make up a small percentage of the enzyme’s surface area (Figure 2). This is a common feature of SABATH methyltransferases [12]. Interestingly, although LAMT exists as a homodimer, dimerization is not necessary for its activity because the amino acids from one LAMT subunit do not form the active center of the other subunit [13].

The catalytic centers of the enzyme are built mainly of polar amino acid residues, which distinguishes LAMT from other SABATH family methyltransferases, in which hydrophobic amino acids predominate. This is due to differences in the structure of substrates subject to methylation. Most of the methyltransferases in this family methylate small, non-polar molecules often containing aromatic rings such as salicylic acid and benzoic acid. However, LAMT binds loganic acid, which contains a hydrophilic glucose moiety and an iridoid ring substituted with polar groups. The diversity in the structure of substrates also results in the size of the binding pocket measured as the solvent-accessible surface, which, in the case of LAMT from Catharanthus roseus, is ~2256 Å^3^ and, for comparison, in SAMT from Clarkia brewerii, is only ~1135 Å^3^ [11].

The aim of this work is to understand and characterize the mechanism of the reaction catalyzed by loganic acid methyltransferase at the molecular level. Considering the important role of loganic acid methyltransferase in the biosynthesis of indole alkaloids, understanding the interaction of the substrate with the enzyme and the mechanism of the catalyzed reaction is crucial from the biotechnological point of view.

The methylation reaction catalyzed by LAMT is a one-step process involving the transfer of a methyl group from the cofactor S-adenosylmethionine to loganic acid. The course of this reaction is an interesting research challenge because the exact mechanism of catalysis is not known. In this work, a detailed characterization of the reaction mechanism was carried out, including determination of the reaction barrier and proposal of amino acid residues in the active site that play a catalytic role. In the first stage, molecular dynamics simulations were conducted, which allowed us to obtain many conformations of the substrate–enzyme complex. Subsequently, the resulting conformations were analyzed to find those that may exhibit catalytic activity. Selected conformations were used to build models of the active site of the enzyme and perform quantum mechanical calculations at the DFT level. The calculations allowed us to obtain the activation energy for the methylation reaction, which enabled quantitative and qualitative comparison of different possible pathways. Finally, the obtained theoretical results were verified by comparison with the available corresponding experimental data.

## 2. Results and Discussion

### 2.1. Review and Initial Analysis of the Previously Published Data

The first step of this work was to select the crystal structure, which was then used to study the catalytic mechanism of loganic acid methyltransferase. Two LAMT structures from *Catharanthus roseus* (PDB codes 6C8S and 6C8R) were found in the RCSB PDB crystal structures database [11]. Both structures were determined by X-ray diffraction and contain LAMT methyltransferase homodimer. Both complexes additionally include S-adenosylhomocysteine, an analog of the S-adenosylmethionine cofactor, which is a methyl group donor in the reaction catalyzed by LAMT. The 6C8S structure was solved with a resolution of 2.20 Å and 6C8R-1.95 Å. The 6C8R structure, unlike the 6C8S, additionally contained loganic acid, forming a complex with LAMT (Figure 3).

In the further stages of the work, solely the 6C8R structure was used because it is characterized by better resolution than the 6C8S structure and also contains both SAH ligands and loganic acid.

Two main conclusions can be drawn from the analysis of the crystal structure of 6C8R. First, the distance between one of the oxygens of the carboxyl group of loganic acid and the sulfur atom of the SAH ligand is quite small (3.3 Å), which is related to the structure of the ligands of SAH, unlike the cofactor SAM, which does not contain a reactive methyl group on the sulfur atom. In the LAMT complex with SAM and loganic acid, steric effects associated with the presence of the methyl group of the cofactor is likely to cause the distance between the cofactor and the substrate to be greater than that in the crystal with SAH. For this reason, it can also be suspected that the conformation of the active site in the crystal structure may deviate from the conformation with the bound cofactor. This hypothesis was tested in this study, as described in the subsequent paragraph. The second observation is related to the way loganic acid is bound by the enzyme. In the SABATH family of methyltransferases, carboxylic acid substrates are bound by a conserved glutamine residue. This residue is also present in the active site of loganic acid methyltransferase but does not form hydrogen bonds with the substrate. However, this interaction is crucial for the course of the reaction because the mutation of this glutamine to alanine leads to a significant reduction in enzyme activity. On this basis, it can be concluded that glutamine 38 is involved in interactions with loganic acid, although this is not observed in the crystal structure of the protein. Such a hypothesis was also suggested in a previous paper [11].

The two above observations lead to the conclusion that the structure of the active center of loganic acid methyltransferase in the 6C8R crystal may significantly differ from that occurring physiologically. In computational studies focused on the analysis of the catalytic mechanism of proteins, application of quantum mechanical calculations is a standard method [14]. However, in the case of loganic acid methyltransferase, performing QM calculations based on the conformation of the protein from the crystal structure may lead to incorrect conclusions. For this reason, an approach combining molecular dynamics and quantum mechanical computation was used in this work. This approach has bee successfully used to study the catalytic mechanism of TrmD methyltransferase, for which the Mg^2+^ cation binding site was unknown [15]. This methodology was also used to characterize the mechanism of NEP1 methyltransferase, for which calculations solely using the crystal structure led to non-physiological reaction barriers [16].

### 2.2. Crystal Structure Analysis and Preparation of MD Simulations

Energy optimization as described in the Materials and Methods section was the first part of this study. The minimized system was then used as the starting geometry for further steps, leading to three independent molecular dynamics trajectories. These trajectories were prepared in a uniform manner, although they differed in the initial velocities of the atoms. This approach enabled us to test whether the properties of the system are reproducible.

In the next step of preparing each of the three trajectories (**1**, **2** and **3**), the system was simulated under NVT conditions for 100 ps in order to achieve thermal stabilization. In the case of LAMT methyltransferase, it was shown that the enzyme is active at 303 K [11]; therefore, the target temperature for each simulation was also 303 K. During the simulation, the heavy atoms of the protein complex were frozen; their position could not change, and water molecules and ions could move freely. The advantage of this approach was that the solvent could fill the available spaces inside the complex while maintaining a structure of the protein similar to that in the crystal. During each of the three NVT simulations, the temperature of the system quickly reached values close to the target temperature of 303 K (Appendix A). After stabilizing the temperature of the system, it was also necessary to stabilize its pressure and, thus, the density of the system. For this reason, simulations were carried out under constant pressure and temperature conditions, with a preserved particle number (NPT) and the heavy atoms of the complex frozen. For each of the three simulations, the system was pressurized to 1 bar (Appendix A).

For most systems, stabilizing the temperature and pressure is sufficient to perform a proper molecular dynamics simulation. In the case of the loganic acid methyltransferase complex studied in this work, additional relaxation was performed under the NPT conditions, with frozen heavy atoms of the main chain of the protein, SAM cofactor and loganic acid, leaving only the side chains relaxed. Analysis of the LAMT crystal structure shows that the interactions between loganic acid and LAMT observed in the crystal may differ from those under physiological conditions. For this reason, additional relaxation, without constraints imposed on the positions of the atoms of the side chains of amino acid residues, allowed for the reorganization of the structure and the formation of additional bonds between the substrate and the protein. This was important because during production simulations, these interactions may be necessary to stabilize the complex. Therefore, subsequent simulations were carried out under NPT conditions, with the bonds removed from the side chains of amino acid residues. The system was simulated for 1000 ps. In all three trajectories, the formation of a hydrogen bond between loganic acid and glutamine 38 was observed, for which a role in substrate binding was postulated [11]. 

In trajectory **1**, this bond was formed between the oxygen atom of the iridoid ring and the amino group of the glutamine residue. This observation is interesting because in other methyltransferases from the SABATH family, this residue is responsible for binding the carboxyl group of the substrate. Nevertheless, it is possible that glutamine 38 has a different function in loganic acid methyltransferase (Figure 4).

In trajectory **2** (Figure 5), however, we observed the formation of a hydrogen bond between the carboxyl group of the loganic acid and the glutamine residue. This way of binding loganic acid is similar to the way that substrates are bound by other methyltransferases in this family.

A similar conclusion can be drawn from the final structure of trajectory **3**. In this trajectory, the carboxyl group of loganic acid also interacts with the side chain of glutamine (Figure 6).

After stabilizing the temperature and pressure of the system, as well as relaxing the side chains of amino acid residues, the next three (production) independent molecular dynamics simulations were performed. The final structures of each of the three trajectories, which are shown in the figures above (Figure 4, Figure 5 and Figure 6), were used as initial structures for production simulations. Although the means of interaction between loganic acid and glutamine 38 in the last step of trajectory **1** are different than in the other two trajectories (**2** and **3**), a production simulation was performed using this structure in order to check whether this bond would be preserved after removing the constrains from the remaining atoms of the complex.

### 2.3. Analysis of Production MD Simulations

After proper preparation of the systems and final snapshots from trajectories **1**, **2** and **3**, production molecular dynamics simulations were performed in order to reproduce the native conformations of the LAMT complex with the cofactor and loganic acid. Three independent trajectories, each with a length of 20 ns, were obtained, which made it possible to study the conformational dynamics of the studied complex. Simulations were carried out in an isobaric–isothermal ensemble, and the previously set constrains on the protein, cofactor and substrate heavy atoms were removed. Then, the following properties of the system were characterized:Stability of the LAMT complex with the SAM cofactor and loganic acid;Interactions that stabilize the binding of loganic acid;The distance between the carboxyl group of loganic acid and the methyl group of the SAM cofactor.

One of the approaches used to determine the stability of the complex during the molecular dynamics simulations is RMSD analysis, which allows provides information on how much the conformation of the complex in a given step of the simulation deviates from the reference structure. The structure from the first step of the simulation was used as a reference. For each of the three trajectories, the RMSD values of Cα carbon atoms of amino acid residues were calculated. For each simulation, the RMSD values remained relatively constant and did not exceed 2.5 Å. There were also no spikes in RMSD values. Relatively constant and low RMSD values indicate that the complex was stable during the molecular dynamics simulation; therefore, the obtained trajectories can be used for further analysis (Figure 7).

In the next step, interactions between loganic acid and loganic acid methyltransferase were analyzed. Hydrogen bonds were determined using the VMD 1.9.3 program. The cutoff values above which atoms were considered not to form hydrogen bonds were 3.2 Å for the donor–acceptor distance and 50° for the angle between the donor, hydrogen and acceptor. Such values allowed for the analysis of strong and medium-strength hydrogen bonds. The results are summarized in Table 1 and shown in Figure 8. 

Several key hydrogen bonds between loganic acid and protein were observed in the molecular dynamics simulations. In all three trajectories, histidine 162 formed a hydrogen bond with the carboxyl group of the substrate. This bond was present for most of the simulation time, which proves a significant role of this residue in the binding of loganic acid. The tryptophan 163 residue also participated in interactions with the carboxyl group of the substrate. However, the frequency of hydrogen bonding between this residue and loganic acid depends, to a large extent, on the trajectory of molecular dynamics. Only in trajectory **2** did this interaction occur during practically the entire simulation, and in trajectories **1** and **3** it was present only in about 5% of the frames. However, the distance between the nitrogen atom in the tryptophan aromatic ring and the oxygen atom of the substrate carboxyl group is relatively short, averaging 4.19 ± 0.42 Å in trajectory 1 and 4.18 ± 0.44 Å in trajectory **3**. A possible explanation for this observation is the presence of weak hydrogen bonds between tryptophan 163 and loganic acid or the occurrence of a hydrogen bond in which the substrate interacts with W163 indirectly with the participation of a water molecule. Water bridge formation was observed in some structures during molecular dynamics simulations. 

The role of histidine 162 and tryptophan 163 in substrate binding is also confirmed by the crystal structure of LAMT, in which both residues formed hydrogen bonds with loganic acid. In addition, the H162A and W163F mutants showed negligible catalytic activity, which also suggests an important role of these amino acid residues in interactions with the carboxyl group of the substrate [11]. 

In the molecular dynamics simulations, glutamine 38 residue was also found to be involved in the bond with the carboxyl group of loganic acid. The interaction involving Q38 occurred all three molecular dynamics trajectories and was present for most of the simulation time. In the crystal structure of LAMT, this bond was absent, but the involvement of Q38 in substrate binding was previously postulated by Petronikolou based on mutational analysis and the conservation of this residue in the SABATH family of methyltransferases [11]. Based on molecular dynamics simulations, it can be concluded that amino acid residues H162, W163 and Q38 participate in the recognition and binding of the carboxyl group of loganic acid and are responsible for the proper orientation of the substrate in order to carry out the methylation reaction. This conclusion is additionally confirmed by previously published experimental studies [11]. 

In the active site of the enzyme, there are also interactions with the sugar moiety of loganic acid. Among others, alanine 241 participates in these interactions, forming a hydrogen bond with the oxygen atom of the glucose hydroxyl group. This binding occurred in each of the performed simulations for about 50% of their duration. Tyrosine 37 was also involved in interactions with the rest of the glucose. However, this binding occurred relatively rarely and was present in only 16.78% of the first trajectory and practically unobserved in the remaining two. For this reason, it can be concluded that the key interaction for the recognition of the glucose ring is the interaction involving alanine 241, while tyrosine 37 plays only a secondary role. 

Both hydrogen bonds were present in the LAMT crystal structure; however, only one of was present for a significant part of the simulation time. Thus, it seems that in the case of Y37 interactions with loganic acid, the hydrophobic interactions between the aromatic ring and the glucose ring play a key role and not hydrogen bonds. The loganic acid methyltransferase mutant Y37A does not show catalytic activity, which further supports the conclusion about the role of tyrosine 37 residue in interactions with the substrate. 

Interactions with the loganic acid iridoid ring also occurred during the molecular dynamics simulation. Glutamine residue 273 participated in them, forming a hydrogen bond with the hydroxyl group of the iridoid ring. Due to the presence of an amide group in the side chain of glutamine 273, this residue can be both a donor and an acceptor of a hydrogen bond; however, in the simulations, the main observed interaction involved the NH_2_ Q273 group. The role of this residue in the recognition of the iridoid ring was previously postulated by Petronikolou [11] based on crystal analysis and mutational studies, further supporting the conclusions of molecular dynamics simulations. 

In the crystal structure, a hydrogen bond was also present between the hydroxyl group in the iridoid ring and the H245 residue, but formation of this bond was not observed in the current simulations. Interestingly, hydrogen bonding involving glutamine 273 occurred most often in trajectory **2** (80.97%) and much less often in trajectories **1** and **3**, as is the case of bonding with tryptophan residue 163. This observation can be explained if we take into account the possibility of formation of hydrogen bonding involving a water molecule between W163 and the substrate. 

The presence of a water bridge was observed in trajectories **1** and **3**, in which Q273 rarely forms a hydrogen bond with the hydroxyl group of the substrate. This is probably because the water molecule forming the water bridge interacts with both the oxygen atom of the carboxyl group (O2) and the oxygen atom of the hydroxyl group of the loganic acid (O9). The steric and electrostatic effects associated with the presence of a water molecule near the hydroxyl group make the hydrogen bond with glutamine 273 weaker and less frequent. The opposite situation can be observed in trajectory **2**, in which direct hydrogen bonding between tryptophan 163 and the substrate is mainly encountered. Due to the lack of a water molecules in the vicinity of O9, the interaction between Q273 and the hydroxyl group occurs more often.

Based on the analysis of hydrogen bonds between loganic acid and the enzyme, it can be concluded that there are two possible patterns of substrate–protein interaction. The first possibility is direct hydrogen bonding between the tryptophan 163 NH group of the indole ring and loganic acid’s carboxylic group, together with strong interaction between glutamine 273 and the iridoid ring. In the second scheme of interactions between W163 and the substrate, there is an indirect hydrogen bond involving a water molecule, and the Q273 residue plays an additional role in the binding of the substrate.

For further characterization of the LAMT complex with loganic acid and the cofactor, the distance between the carbon atom of the reactive methyl group of S-adenosylmethionine and the methylated oxygen atom of the substrate was calculated. This distance is of great importance from the point of view of the reaction catalyzed by the methyltransferase because it corresponds approximately to the reaction coordinate. For large distances, the methyl group must first approach one of the oxygen atoms of the carboxyl group of the loganic acid, and only then can the methyl group be transferred. For this reason, the reaction barrier is associated not only with the breaking and formation of chemical bonds but also with conformational changes within S-adenosylmethionine. For this reason, short distances between the methyl group and its acceptor are more conducive to the methylation reaction. During each trajectory, the distance between the carbon atom of the methyl group of SAM and the nearest oxygen atom of the substrate carboxyl group is stable and rarely exceeds 3.4 Å (Figure 9). This observation additionally supports the conclusion about the stability of the simulations of molecular dynamics and the SAM–loganic acid interactions occurring therein.

### 2.4. System Preparation for QM Calculations

Molecular dynamics is a method that provides a large amount of information about the conformational changes of the complex. However, it is impossible to analyze the reaction mechanism for each structure obtained as a result of the simulation. For this reason, an analysis of the obtained conformations was carried out in order to find those that would potentially promote catalysis and would differ in the geometry of the active site. The main criterion for the selection of structures was the SAM–loganic acid distance due to its importance in the methylation reaction. Only those structures in which this distance was below 2.8 Å were used for further analysis. From all conformations of the complex satisfying the given cutoff condition, representative structures were then selected for each distinct configuration of the active site. In this way, two different structures were selected that corresponded to different substrate–protein interaction patterns. These structures were then used for QM calculations in order to study the influence of the active site conformation on the activation energy and to characterize the reaction mechanism.

The structure of the active site in both selected frames from the molecular dynamics simulation is quite similar. In both frames, loganic acid interacts with histidine 162 and glutamine 38, which guarantee not only stable binding of loganic acid but also correct orientation of its carboxyl group. The major difference between those two structures is the way the substrate interacts with tryptophan 163. In the first structure (**model 1**), W163 interacts indirectly with loganic acid. In this structure, there is a hydrogen bond involving a water molecule that interacts not only with the carboxyl group of the substrate but also with the hydroxyl group of the iridoid ring (Figure 10).

In the second structure (**model 2**), it can be observed that the hydrogen bond between loganic acid and tryptophan 163 occurs without the participation of a water molecule. W163 is located in the immediate vicinity of the carboxyl group of the substrate and interacts directly via the amino hydrogen atom of its indole ring with the oxygen atom of loganic acid’s carboxyl group (Figure 11).

MD simulations allowed us to study the dynamics of the active center of loganic acid methyltransferase and to reconstruct the native conformations of the complex. Based on the simulations, the key hydrogen bonds between the substrate and the enzyme were characterized, allowing us to select the most representative frames from the simulation in which these interactions were preserved. These structures were used in the next step, as presented below, to study the mechanism of the reaction catalyzed by LAMT.

### 2.5. Catalytic Mechanism Analysis

A quantum chemical cluster approach was used to analyze the mechanism of the reaction catalyzed by loganic acid methyltransferase. In this method, a relatively small but well-selected part of the enzyme is cut out from the structure of the studied complex and is treated at the quantum chemical level. This approach was successfully used in previous studies devoted to the analysis of mechanisms of reactions catalyzed by enzymes [15,17,18,19]. 

In the first stage of quantum mechanical calculations, two models (**model 1** and **model 2**) of the active center were prepared using previously selected structures from molecular dynamics simulations. In both models, amino acid residues located in the immediate vicinity of the methyl group of SAM and the carboxyl group of loganic acid, including residues forming key interactions, were taken into account. Then, the atoms at the truncated point were fixed to prevent their unnatural movement during the optimization of the geometry. These models contained the same amino acid residues, loganic acid and S-adenosylmethionine. Exactly the same atoms were fixed in both models. The total number of atoms in **model 1** was 192, and that in **model 2** was 189. The main difference between the models, the influence of which was examined during the course of the reaction, was the presence of a water molecule in **model 1**, resulting in the number of atoms in this model increasing by three compared with **model 2**. This water molecule participated in the hydrogen bonding between W163 and the substrate (Appendix A). In **model 2**, this binding was direct (Appendix A). 

After building the models, it was necessary to optimize their geometry in order to find the energy minimum corresponding to the system before the methylation reaction. The DFT method was used for the calculations, which allows for an accurate description of the energy at a relatively low computational cost. Correlation-exchange functional B3LYP [20,21,22,23,24] was used, which is widely applied in studies of the mechanisms of enzymatic reactions using the cluster approach [25]. However, this functional is not able to satisfactorily describe dispersion interactions that are crucial for the intermolecular interaction; therefore, empirical correction GD3BJ for dispersion forces was included in the calculations [26]. The Def2-SVP basis set was used for geometry optimization, which is recommended for calculations of similar systems using DFT methods [27]. The charge of both systems was set to 0, and the spin was set to the singlet state. In order to confirm whether the identified geometries corresponded to the actual energy minimum, hessians were computed, showing only a few imaginary frequencies associated with fixed atoms, which were of the order < 30i cm^−1^.

After optimizing the geometry of the systems, we checked whether the obtained structures differed significantly from those obtained before energy minimization. For this purpose, the structures from the first and last optimization step were superimposed. No major changes in the conformations of either **model 1** or **model 2** were observed, which means that the applied atom constraint scheme allowed for the reproduction of steric effects of the part of the complex not included in the model (Appendix A).

The optimized structures for **models 1** and **2** were used to find the transition state for the methylation reaction of loganic acid involving S-adenosylmethionine. In order to find the approximate structure of the transition state, the relaxed potential energy surface scan method was used. This method makes it possible to determine changes in potential energy during the change of one of the geometrical parameters. In both models, the distance between the carbon atom of the methyl group of SAM and the oxygen of the carboxyl group of loganic acid was scanned. During each individual calculation, all degrees of freedom were optimized, except for the selected distance, which remained constant. The reaction coordinate was scanned every 0.2 Å. For both models, the maximum electron energy corresponds to distance of 2.1 Å between the methyl group of the cofactor and its acceptor (Table 2).

The most distant energy structures (distance 2.1 Å) from the scans were used to find the transition state for the methyl group transfer reaction. First, the hessians were calculated to confirm that they were close to the transition state. Both structures showed one imaginary frequency with a high value, which corresponded to the vibrations associated with the methyl group. The presence of the imaginary vibration frequency confirmed that the geometry of these structures is close to the geometry of the transition state, so they can be used as a starting point for its optimization. The transition states were optimized using the analytical hessian calculated in the previous step. Then, it was confirmed that the obtained structures corresponded to the saddle point by calculating the eigenvalues.

In the transition states for both models, the methyl group is located between the sulfur atom of S-adenosylmethionine and the oxygen of loganic acid, which is its acceptor. The S-CH_3_-O angle is 174° in **model 1** and 170° in **model 2**, which means that the methyl group approaches the acceptor in the plane defined by the carboxyl group. The S-CH_3_ distance in both models is 2.3 Å, and the CH_3_-O distance is slightly shorter in **model 1** than in **model 2** (2.1 Å vs. 2.2 Å). The asymmetric methyl transfer transition state is characteristic for many methyltransferases, including salicylic acid methyltransferase, which, like LAMT, belong to the SABATH family of methyltransferases. The corresponding distances for the transition state of the reaction catalyzed by SAMT are 2.4 Å and 2.0 Å [28]. In the transition state for both models, the methyl group adopts a flat conformation, and the reaction itself proceeds with an inversion of configuration on the carbon atom. In both transition states, the interactions present in the system before the reaction are preserved. The carboxyl group of the substrate is oriented to accept the methyl group through interactions with glutamine 38, histidine 162 and tryptophan 163 (Appendix A).

After modeling both the initial structures and the transition state structures, the structures of the final state were determined. Optimized conformations from a scan for a CH_3_-O distance of 1.9 Å were used for this purpose. The constraints imposed on the distance between the methyl group and the acceptor were removed, and the geometry of the system was optimized. Then, it was verified whether the system was an energy minimum by counting the eigenvalues. In both models, as a result of a chemical reaction, the methyl group of the cofactor was transferred to the carboxyl group of loganic acid, resulting in the formation of a methyl ester–loganin complex. 

The hydrogen bonds involving the loganin ester group differ between models. In **model 1**, the same interactions that were responsible for the orientation of the carboxyl group of loganic acid are preserved. The ester group is bound with the participation of glutamine 38, which interacts with the oxygen atom substituted with a methyl group. The loganin carbonyl group forms a direct hydrogen bond with histidine 162 and a hydrogen bond with tryptophan 163 involving a water molecule. In **model 2**, the direct interactions between the carbonyl group of loganin with tryptophan 163 and histidine 162 are preserved. As a result of the transfer of the methyl group, the hydrogen bond involving glutamine 38 is broken. During the methyl group transfer reaction, the carboxyl group of loganic acid loses its negative charge, which may significantly weaken the hydrogen bond with Q38. In addition, the presence of a bulky group near the oxygen atom may further weaken this bond due to steric hindrance. The combination of electrostatic and steric effects can lead to a strong weakening of the hydrogen bond between loganin and glutamine 38, resulting in its breaking. The lack of interaction between Q38 and the reaction product may be the basis for the mechanism of loganin dissociation from the active site of loganic acid methyltransferase in order to start a new catalytic cycle.

After determining the structure of stationary points, energy corrections were calculated for both models. In order to obtain more accurate values of the electronic energy, single-point calculations were performed using the Def2-QZVP basis set. In the case of DFT methods, the use of a larger basis set than Def2-QZVP does not lead to a further increase in the accuracy of the results due to the basis set limit [27]. The solvation energy was also calculated using the CPCM implicit model with two different dielectric constants (4 and 80) [29,30]. System vibrations at 0 K were taken into account by computing zero-point energy (ZPE). The final energy is the sum of the electronic energy calculated using the Def2-QZVP basis set, the solvation energy (ε = 4 or 80) and the zero-point vibrational energy.

In both models, increasing the basis set from Def2-SVP to Def2-QZVP leads to an increase in the relative electron energy of the transition state and the product by several kcal/mol, which means that some aspects of the electron density are poorly described in calculations using a smaller basis set. In both **model 1** and **model 2**, the absolute value of the solvation energy decreases in the following order: initial state, transition state, final state. During the chemical reaction, the positively charged S-adenosylmethionine and the negatively charged loganic acid are converted into neutral products. Due to the fact that ionic substances are more strongly solvated, a decrease in solvation energy is observed during the reaction. The calculated zero-point vibrational energy represents only a small contribution to the final energy of the system: of the order of 1 kcal/mol. In both models, this correction increases the final state energy, and in **model 2**, the zero-point vibrational energy also lowers the reaction barrier (Table 3).

Corrections to the electronic energy allowed for the determination of the energy profile of the reaction (Figure 12 and Figure 13). In both models, the reaction barrier is related to the transfer of the methyl group from S-adenosylmethionine to the carboxyl group of loganic acid. The reaction barriers for both models are similar. In the case of **model 1**, the activation energy is 18.7 kcal/mol, and in **model 2**, it is 17.6 kcal/mol. In both models, the reaction is exoenergetic.

In order to check the influence of the choice of dielectric constant on the reaction barrier, the solvation energy was calculated using two diverse dielectric constants (ε = 4 and 80). Changing the value of the dielectric constant from 4 to 80 increased both the reaction barrier and the final state energy. However, these differences are so small that the choice of dielectric constant does not significantly affect the interpretation of the obtained energy profiles. This means that most of the significant electrostatic interactions were taken into account during the construction of both models.

Analysis of the reaction pathways using quantum chemistry methods and a cluster approach enabled the characterization of two probable reaction mechanisms. In both mechanisms, the S methyl group of adenosylmethionine is transferred to the carboxyl group of loganic acid, resulting in S-adenosylhomocysteine and loganin. This process is the rate-limiting stage of the reaction. The difference between these mechanisms is how the tryptophan 163 residue interacts with loganic acid. In the first mechanism, the hydrogen bond between the W163 residue and the substrate occurs with the participation of a water molecule, while in the second mechanism, the bond is direct. Reaction barriers for these two possibilities are close to each other and equal 18.7 and 17.6 kcal/mol, respectively, which are adequate values for reactions catalyzed in living organisms. This means that both active site configurations of loganic acid methyltransferase are catalytically active.

The obtained results also allow for a deeper understanding of some experimental facts. Earlier experimental studies showed that the turnover number (k_cat_) for LAMT is 0.31 ± 0.01 s^−1^, which corresponds to an energy barrier of 18.5 kcal/mol [11]. This barrier is related to the slowest step after binding of substrates, which may be product release, conformational change or chemical reaction. This value is very close to the activation energy for the reaction mechanisms characterized in this work (18.7 kcal/mol for **model 1** and 17.6 kcal/mol for **model 2**). This allows us to interpret the k_cat_ constant as the rate constant of the reaction of the methyl group transfer from S-adenosylmethionine to loganic acid and the reaction itself as the limiting step in the entire catalytic cycle. The agreement between the experimental results and the quantum mechanical calculations is additionally confirmed by the catalytic mechanism characterized in this work.

## 3. Materials and Methods

### 3.1. Structural Preparation

Fully atomistic molecular dynamics simulations were performed using the GROMACS package (version 2016.5) [31]. A crystal structure of from the RCSB database (PDB code 6C8R) [11] was used as the starting structure, which contained a complex of loganic acid methyltransferase with loganic acid and SAH, which was replaced in the simulation with S-adenosylmethionine. 

The crystal structure was protonated to pH 7.5 using the PDB2PQR server [32]. The protein atoms and the SAM cofactor were described using the CHARMM36 force field [33], and the water molecules were described with the TIP3P solvent model [34]. The missing parameters for loganic acid were generated by the CHARMM GUI server using the Ligand Reader and Modeler tool, then converted to GROMACS format using the Force Field Converter tool [35,36]. Na^+^ cations and Cl^−^ anions were added to the system to neutralize the protein charge and to obtain a NaCl concentration of 0.15 M. Energy minimization of the system was carried out using the steepest descent algorithm until the maximum force acting on the system was <1000 kJ mol^−1^ nm^−1^. 

### 3.2. Molecular Dynamics Simulations

The minimized system was equilibrated under NVT conditions at 303 K for 100 ps using a modified Barendsen thermostat (V-rescale). The system was then simulated for 100 ps in the NPT ensemble. A constant pressure of 1 bar was maintained by a Parrinello–Rahman barostat. In both relaxation steps, the heavy atoms of protein, SAM and loganic acid were frozen. In the next step, bonded relaxation was performed on the heavy atoms of the protein backbone, SAM and loganic acid. The system was simulated under NPT conditions for 1 ns. Such a balanced system was subjected to proper dynamics in the NPT ensemble for 20 ns. The Lennard Jones potential was used to describe the van der Waals interactions, and the Coulomb potential was used to describe the electrostatic interactions. The values of cutoff radii for non-bonding interactions were equal and amounted to 1.2 nm. Long-range electrostatic interactions were taken into account using the PME method. The use of the LINCS algorithm [37] enabled the use of a time step of 2 fs.

### 3.3. Quantum Mechanical Calculations

All quantum mechanical calculations were performed using Gaussian16 software [38]. Cluster models were built based on structures from molecular dynamics simulations. Geometry optimization was performed using the B3LYP functional [20,21,22,23,24], taking into account empirical correction for GD3BJ dispersion [26]. Atoms on the periphery of the model were frozen during optimization. Def2 SVP [27] was used as the basis set. At the same level of theory, the ZPE energy correction and the solvation energy were calculated using the CPCM continuous solvent model for two different dielectric constants: −4 and 80 [29,30]. In order to obtain more accurate values of the electron energy for stationary points, single-point calculations were also performed using the Def2 QZVP basis set.

### 3.4. Processing and Analysis Details

Analysis of the trajectories obtained from molecular dynamics simulation was performed using GROMACS 2016.5, VMD 1.9.3 and PYMOL 2.5.2 programs. Structures for QM calculations were prepared using the GaussView 6 program. Python 3 scripts were used to prepare the graphs.

## 4. Conclusions

Molecular dynamics simulations enabled the reconstruction of the native conformations of loganic acid methyltransferase and the characterization of interactions between the substrate and the enzyme. In the simulations, the structure of the active site of the enzyme was reorganized. It was observed that glutamine 38 forms a hydrogen bond with the carboxyl group of loganic acid due to a change in the conformation of the side chain. The role of Q38 in substrate binding was postulated in earlier studies focusing on the analysis of the LAMT crystal structure and mutations. Petronikolou et al. [11] also suggested that H162 and W163 residues are involved in interactions with the carboxyl group of the substrate, in addition to Q38, which was confirmed in our work.

During the molecular dynamics simulations, the double role of tryptophan 163 was also identified. This residue participates not only in the formation of a direct hydrogen bond, as observed in the LAMT crystal structure, but also in the formation of an indirect bond involving a water bridge, the carboxyl group of the substrate and the hydroxyl group of the iridoid ring. Therefore, tryptophan 163 is not only responsible for the correct orientation and binding of loganic acid but may also play an additional role in the recognition of the substituted iridoid ring.

The performed quantum mechanical calculations for structures corresponding to different interaction patterns between the substrate and the enzyme showed that both configurations of the active site favor the methyl group transfer reaction. The height of the obtained reaction barriers is consistent with the predictions for the reaction taking place in living organisms. Comparison of reaction barriers determined using QM calculations with the barrier of the slowest process following substrate binding by LAMT allowed us to interpret the experimental k_cat_ constant as a rate constant for the methyl group transfer reaction [11]. This process is the rate-limiting stage of the entire catalytic cycle.

The combined MD and QM approach used in this work allowed us not only to reproduce experimental observations but also provided new details on the catalytic mechanism of loganic acid methyltransferase. Understanding the molecular basis of the recognition of the glycosylated and hydroxylated iridoid ring of the substrate by loganic acid methyltransferase may allow for the design of new methyltransferases with altered regioselectivity or substrate specificity. These methyltransferases can be used in the pharmaceutical industry for the biosynthesis of drug precursors and as safe biocatalysts in alkylation reactions. This work is also important from the methodological point of view, as it shows the advantages of a hybrid approach combining molecular dynamics and QM calculations. In particular, the methodology used here identified the role of the water bridge in the LAMT catalytic mechanism, which would have been difficult to detect using experimental techniques.

## Figures and Tables

**Figure 1 molecules-28-05767-f001:**
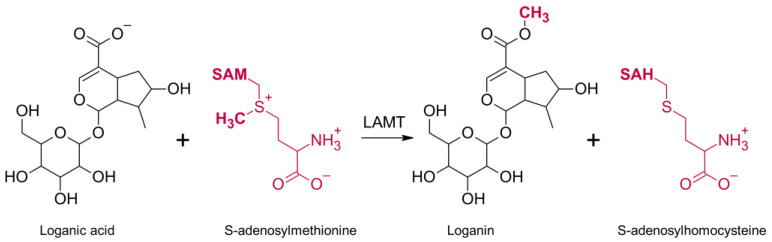
The mechanism of the reaction catalyzed by loganic acid methyltransferase (LAMT). Loganic acid is methylated on the oxygen atom of the carboxyl group and converted to loganin. At the same time, the methyl group donor, S-adenosylmethionine, turns into S-adenosylhomocysteine.

**Figure 2 molecules-28-05767-f002:**
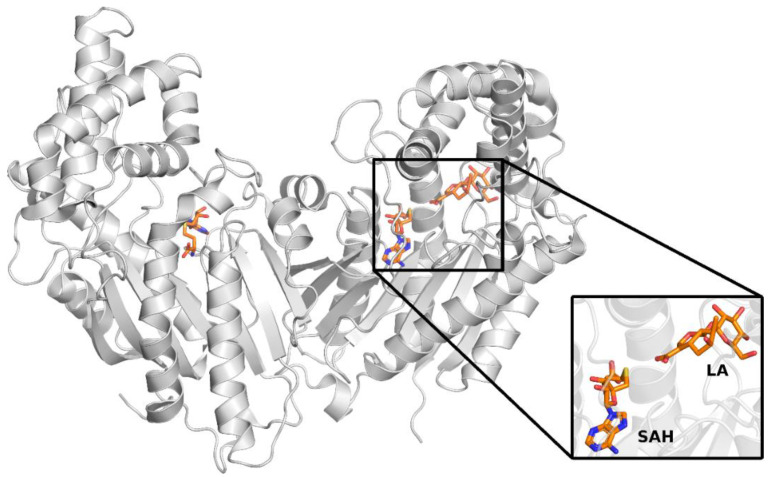
Crystal structure of loganic acid methyltransferase (PDB code 6C8R). The black insert shows a close-up of the enzyme’s active site, where S-adenosylhomocysteine (SAH) and loganic acid (LA) are bound.

**Figure 3 molecules-28-05767-f003:**
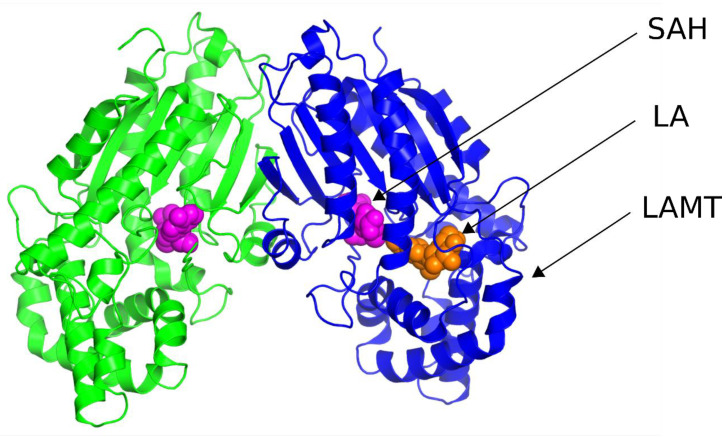
Crystal structure of loganic acid methyltransferase (LAMT) (PDB code 6C8R). Particular LAMT subunits are marked in green and blue. The SAH ligand (pink) and loganic acid (LA) (orange) atoms are represented by Van der Waals spheres.

**Figure 4 molecules-28-05767-f004:**
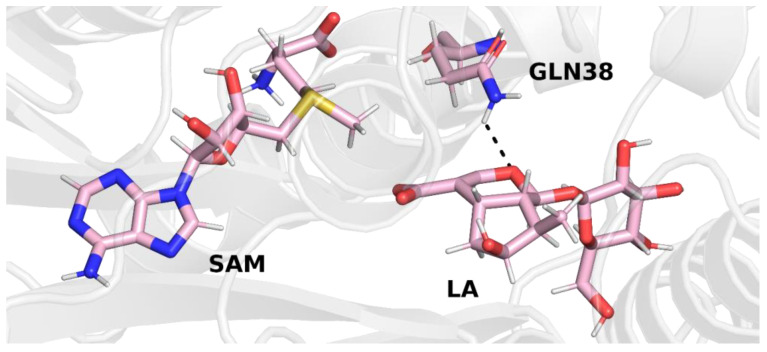
Final structure from trajectory **1**. Glutamine 38 forms a hydrogen bond with the oxygen atom of the iridoid ring of loganic acid.

**Figure 5 molecules-28-05767-f005:**
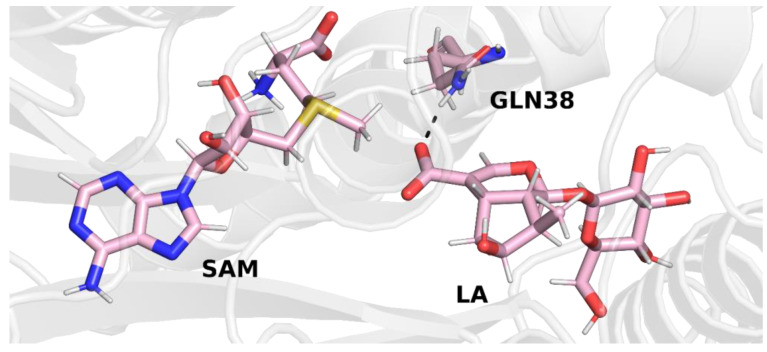
Final structure from trajectory **2**. Glutamine 38 forms a hydrogen bond with the carboxyl group of loganic acid.

**Figure 6 molecules-28-05767-f006:**
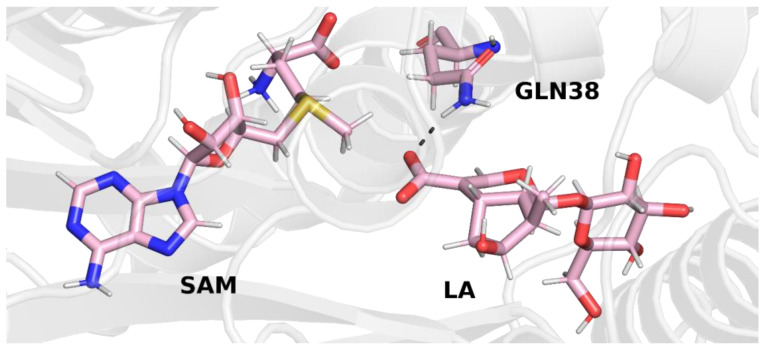
Final structure from trajectory **3**. Glutamine 38 forms a hydrogen bond with the carboxyl group of loganic acid.

**Figure 7 molecules-28-05767-f007:**
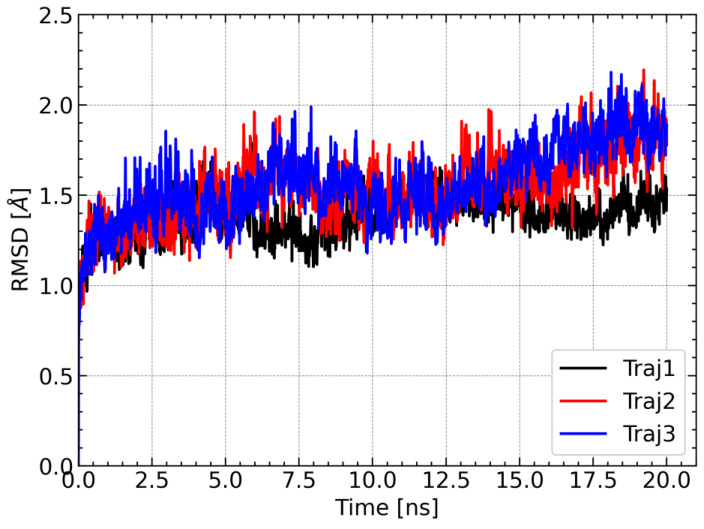
Changes in RMSD values of Cα carbons of amino acid residues during each of the three molecular dynamics trajectories. The structure from the first step of the simulation was used as a reference structure to calculate the RMSD.

**Figure 8 molecules-28-05767-f008:**
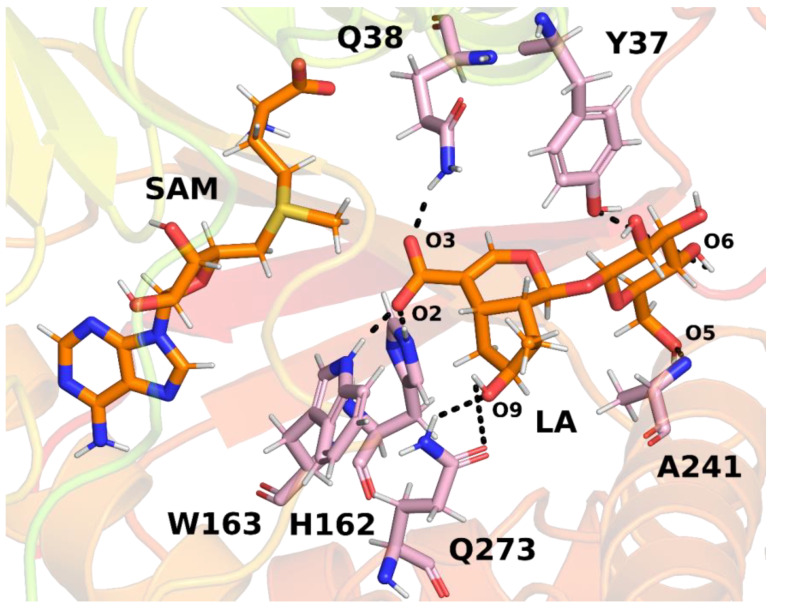
The most common hydrogen bonds between loganic acid and protein based on an example structure from the simulation. Bonds between specific atoms are marked with dashed lines.

**Figure 9 molecules-28-05767-f009:**
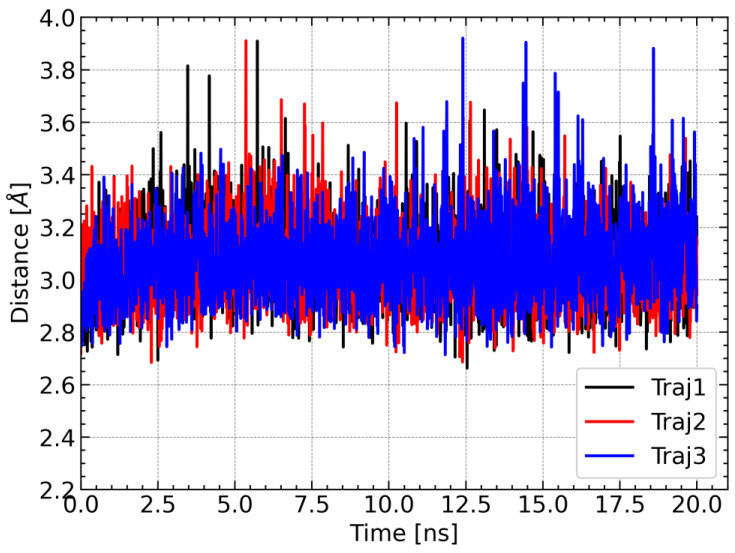
Changes in the distance between the carbon atom of the methyl group of SAM and the nearest oxygen atom of the carboxyl group of loganic acid during the simulation.

**Figure 10 molecules-28-05767-f010:**
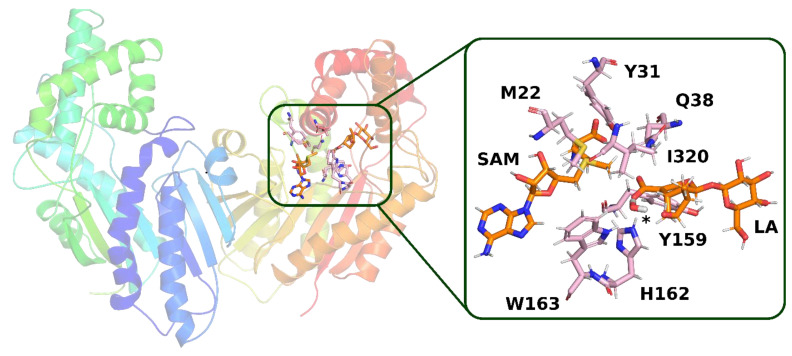
Structure from molecular dynamics (**model 1**) selected for QM calculations. In the active site of the enzyme, a hydrogen bond is visible with the participation of a water molecule, the position of which is marked with an asterisk. Amino acid residues, which, together with loganic acid and S-adenosylmethionine, were included in the QM calculations, are labeled.

**Figure 11 molecules-28-05767-f011:**
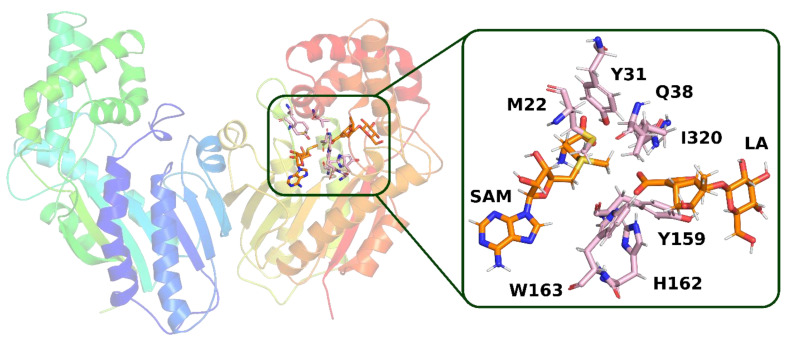
Structure from molecular dynamics (**model 2**) selected for QM calculations. In the active site of the enzyme, a direct hydrogen bond between loganic acid and tryptophan 163 is present. Amino acid residues, which, together with loganic acid and S-adenosylmethionine, were included in the quantum mechanical calculations, are labeled.

**Figure 12 molecules-28-05767-f012:**
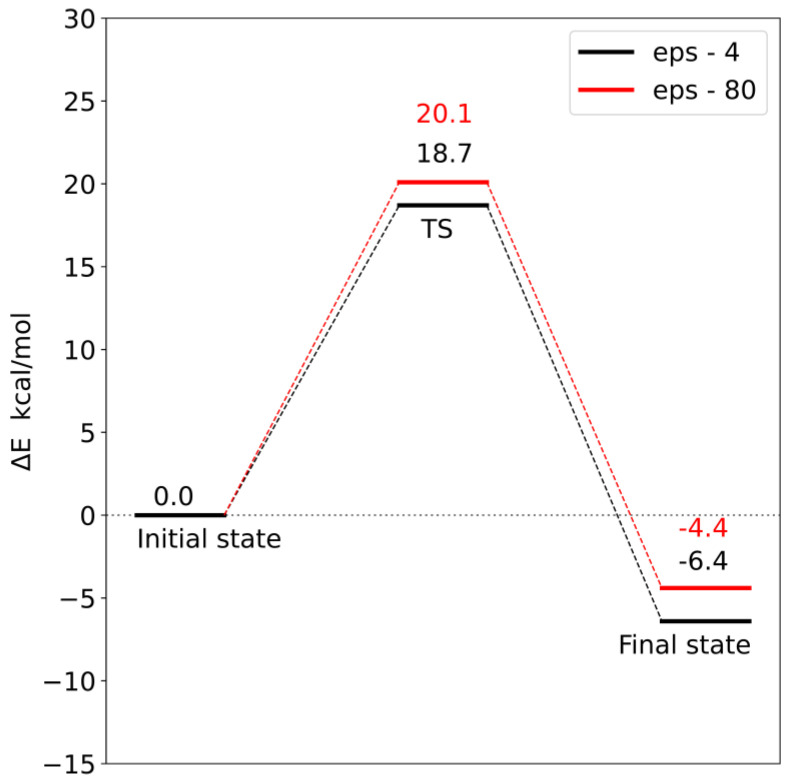
Energy profile for **model 1**. TS—transition state.

**Figure 13 molecules-28-05767-f013:**
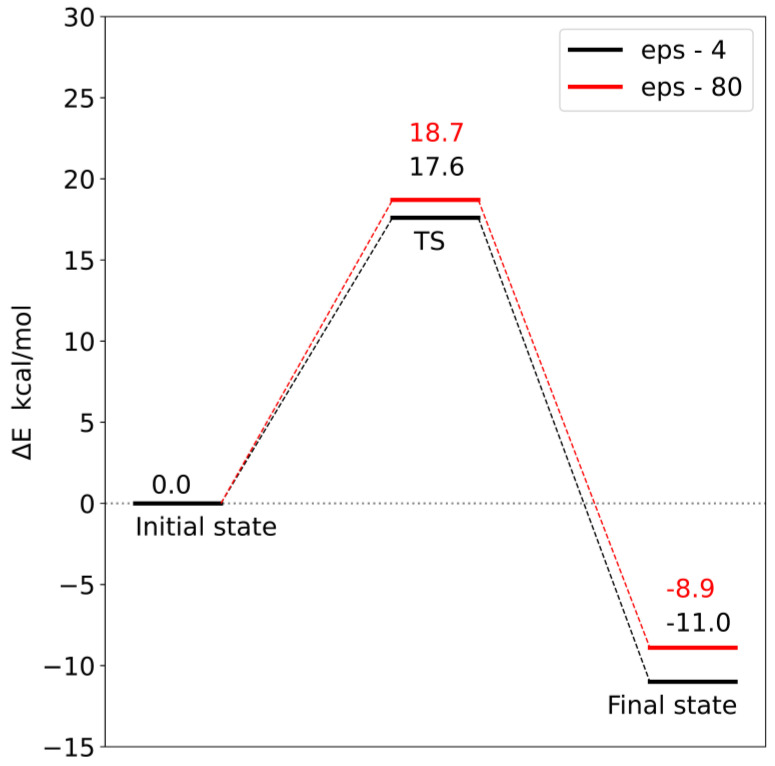
Energy profile for **model 2**. TS—transition state.

**Table 1 molecules-28-05767-t001:** Frequency of occurrence of specific hydrogen bonds between loganic acid and LAMT. The results are shown only for bonds that were present more than 5% of the time during the simulation in at least one of the three trajectories. An explanation of the names of loganic acid atoms is given in the figure above (Figure 8).

Hydrogen Bond		Occupancy (%)		
Donor	Acceptor	Trajectory 1	Trajectory 2	Trajectory 3
H162 side chain	LA-O2	94.91%	92.76%	89.71%
Q38 side chain	LA-O3	96.30%	92.46%	94.46%
Q273 side chain	LA-O9	20.18%	80.97%	15.53%
W163 side chain	LA-O2	5.34%	74.28%	5.29%
A241 backbone	LA-O5	43.76%	60.84%	55.39%
LA-O9	Q273 side chain	7.94%	0.90%	15.33%
LA-O6	Y37 side chain	16.78%	0.30%	0.05%

**Table 2 molecules-28-05767-t002:** Relaxed potential energy scan for the distance between the methyl group of S-adenosylmethionine and the oxygen atom of the carboxyl group of loganic acid. The energy values are relative to the energy of the substrate, for which a value of 0 kcal/mol was assumed.

	**Relative Electronic Energy (kcal/mol)**	
**Distance (Å)**	**Model 1**	**Model 2**
2.7	0.6	1.4
2.5	2.4	4.6
2.3	6.9	10.2
2.1	12.5	12.0
1.9	3.8	3.7

**Table 3 molecules-28-05767-t003:** Values of individual contributions to the final energy of the system expressed in kcal/mol. ΔE_MB_—relative electronic energy calculated in the Def2-SVP basis; ΔE_LB_—relative electronic energy calculated in the Def2-QZVP basis; E_solv_ (ε = 4)—solvation energy calculated with a dielectric constant equal to 4; E_solv_ (ε = 80)–solvation energy calculated with a dielectric constant equal to 80; ΔZPE–relative zero-point vibrational energy.

		ΔE_MB_	ΔE_LB_	E_solv_ (ε = 4)	E_solv_ (ε = 80)	ΔZPE
Model 1	Initial state	0.0	0.0	−33.3	−45.7	0.0
	TS	12.7	15.0	−29.7	−40.7	0.1
	Final state	−18.6	−13.2	−28.2	−38.6	1.7
Model 2	Initial state	0.0	0.0	−32.6	−44.6	0.0
	TS	13.3	15.1	−29.2	−40.0	−1.0
	Final state	−22.2	−18.2	−27.0	−36.9	1.56

## Data Availability

The raw data can be obtained from the corresponding author (M.J.) by email request.

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
