# Peer review of "The Reaction Mechanism of Loganic Acid Methyltransferase: A Molecular Dynamics Simulation and Quantum Mechanics Study"

_molecules, 2023, doi:10.3390/molecules28155767_

Round 1

Reviewer 1 Report

In the manuscript (ID: molecules-2530656), the catalytic mechanism of loganic acid methyltransferase was characterized using molecular dynamics and quantum mechanics methods.

The authors performed molecular dynamics simulations each with length of 20ns. However, the molecular dynamics simulations should be longer about 100 ns and it should be discussed in the manuscript. The description of the table 2 is on the other page in the manuscript.

Author Response

Comment:

In the manuscript (ID: molecules-2530656), the catalytic mechanism of loganic acid methyltransferase was characterized using molecular dynamics and quantum mechanics methods.

The authors performed molecular dynamics simulations each with length of 20ns. However, the molecular dynamics simulations should be longer about 100 ns and it should be discussed in the manuscript. The description of the table 2 is on the other page in the manuscript.

Response:

Thank you very much for the significant effort needed to create this review. We’ve found your suggestions very helpful in improving the quality of our manuscript. Below, please find the direct responses to your comments.

We agree with the Reviewer that in many of the studies using molecular dynamics simulations, longer production runs (about 100 ns) are usually employed. However, we would like to emphasize that in our study we haven’t analyzed longer trajectories due to the multiple reasons, presented below.

Firstly, the main aim of this study was to the analyze the mechanism of the reaction catalyzed by loganic acid methyltransferase, with particular emphasis on reaction pathway, at the quantum mechanical level. Therefore, the molecular dynamics simulations serve mainly as an initial tool to prepare the system for the detailed QM analysis, restoring its physiological state in solution.

To achieve this natural state, the main requirement was to reorient the backbones, and create the hydrogen bonding network with water, which usually requires the time of picoseconds. Our MD simulations lasted tens of nanoseconds, therefore were over thousand times longer than the required ones.

Also, it is a common practice in similar studies using the hybrid MM/QM approach to study the mechanism of enzymatic reactions that the researchers don’t employ the MD simulations at all, using solely the results from molecular docking. Therefore, we’ve gone even further by employing the MD to create the model with the highest possible similarity to the experimental system.

Besides, even in the most advanced and comprehensive recent studies, presenting the results obtained using the methods similar to the ones from our work, the authors have used MD simulations of 20 ns, which is exactly the same value as in our work [10.1002/chem.202104167 ; 10.1016/j.csbj.2019.06.016 ; 10.3389/fchem.2018.00606 ].

The description of the Table 2 has been moved, so it is now on the same page as the Table 2.

Reviewer 2 Report

The manuscript titled 'The reaction mechanism of loganic acid methyltransferase: a molecular dynamics simulation and quantum mechanics study' by Dariusz Maciej Pisklak and co-authors reports the results of MD and DFT calculations devoted to studying the mechanisms of methylation occurring in living organisms. The introduction provides the basics of the object of the study, and all methods are adequately selected, with an appropriate design of the models.

I have some suggestions for the authors to help clarify the text:

  1. Lines 119-166, which are part of the 'results and discussion' section, give a description of experimentally resolved crystal structures made by other authors and published in the literature. The authors make some suggestions and preliminary conclusions and justify the selection of the appropriate model. Here, I see a contradiction. Describing somebody else's results cannot be placed in the 'own results' section. Please put these considerations in a separate subchapter.

  2. In Figures 4-6 and their captions, the authors wrote 'bond,' but they definitely meant a hydrogen bond. Please correct this in the captions.

  3. For one of the systems, the authors highlighted the importance of bridging water molecules in some intermediate complexes during the reaction. Are these molecules taken into account on the DFT level?

Author Response

Comment:

The manuscript titled 'The reaction mechanism of loganic acid methyltransferase: a molecular dynamics simulation and quantum mechanics study' by Dariusz Maciej Pisklak and co-authors reports the results of MD and DFT calculations devoted to studying the mechanisms of methylation occurring in living organisms. The introduction provides the basics of the object of the study, and all methods are adequately selected, with an appropriate design of the models.

I have some suggestions for the authors to help clarify the text:

Response:

Thank you very much for the significant effort needed to create this review. We’ve found all of your suggestions very helpful in improving the quality of our manuscript. Below, please find the direct responses to your comments.

Comment:

Lines 119-166, which are part of the 'results and discussion' section, give a description of experimentally resolved crystal structures made by other authors and published in the literature. The authors make some suggestions and preliminary conclusions and justify the selection of the appropriate model. Here, I see a contradiction. Describing somebody else's results cannot be placed in the 'own results' section. Please put these considerations in a separate subchapter.

Response:

We agree with the Reviewer. We’ve corrected this by creating a separate subchapter entitled “Review and initial analysis of the previously published data” that now contains the text previously present in lines 119-166.

Comment:

In Figures 4-6 and their captions, the authors wrote 'bond,' but they definitely meant a hydrogen bond. Please correct this in the captions.

Response:

Of course, this was an editorial mistake and it has been corrected, as suggested by Reviewer, by replacing “bond” with “hydrogen bond” in each case (captions of Figures 4-6).

Comment:

For one of the systems, the authors highlighted the importance of bridging water molecules in some intermediate complexes during the reaction. Are these molecules taken into account on the DFT level?

Response:

Of course, presence of water molecule is crucial for the system named Model 1. This system, subjected to QM (DFT) studies, contains the bridging water molecule. This was now stated clearly in the Figure 10 caption “ (…) In the active site of the enzyme, a hydrogen bond is visible with the participation of a water molecule, the position of which is marked with an asterisk. (…)”

Round 2

Reviewer 1 Report

The authors answer for my comment appropriately.